# Chain Of Thought Prompting Under Streaming Batch: A Case Study

**Yuxin Tang**
Department of Computer Science
Rice University
Houston, TX 77005, USA
{yuxin.tang}@rice.edu

## Abstract

Recently, Large Language Models (LLMs) have demonstrated remarkable capabilities. Chain-of-Thought (CoT) has been proposed as a way of assisting LLMs in performing complex reasoning. However, developing effective prompts can be a challenging and labor-intensive task. Many studies come out of some way to automatically construct CoT from test data. Most of them assume that all test data is visible before testing and only select a small subset to generate rationales, which is an unrealistic assumption. In this paper, we present a case study on how to construct and optimize chain-of-thought prompting using batch data in streaming settings.

## 1 Introduction

Large Language Models (LLMs) have shown emergent abilities. Wei et al. (2022a) Wei et al. (2022b) discover Chain-of-Thought (CoT) prompting as a simple and broadly applicable method for enhancing reasoning in language models. Many work Zhou et al. (2022a); Wang et al. (2022b); Shi et al. (2022); Zhang et al. (2023a;b); Wang et al. (2022a); Zhou et al. (2022b); Fei et al. (2023); Yang et al. (2023); Shi et al. (2023); Diao et al. (2023) have tried to make further improvements based on CoT.

## 2 Problem Statement

The problem of *chain-of-thought under streaming* is first proposed by Zhang et al. (2023a). It assumes that the test dataset, denoted as $D$, will be evenly partitioned into $m$ batches, each containing $N$ samples. These batches are fed into a large language model $M$ in a continuous stream-like manner. At each time step $t_k$ for $k = 1, 2, \ldots, m$, $M$ processes one batch of questions $\left\{ q_1^{(k)}, q_2^{(k)}, \ldots, q_N^{(k)} \right\}$ using the same prompt $P$. After processing a batch, rationales generated by $M$ are denoted as $\left\{ c_1^{(k)}, c_2^{(k)}, \ldots, c_N^{(k)} \right\}$. A *prompting optimization function* $f(P|(q_1^{(k)}||c_1^{(k)}), (q_2^{(k)}||c_2^{(k)}), \ldots, (q_N^{(k)}||c_N^{(k)}))$[1] is applied to update prompt $P$ before processing the next batch. This allows the model to maintain a coherent chain-of-thought within each batch in the stream, known as the *intra-batch chain-of-thought*.

Function $f$ adopted in Zhang et al. (2023a) is a simple concatenation function by appending all the newly generated question-rationale pairs $(q_1^{(k)}||c_1^{(k)}), (q_2^{(k)}||c_2^{(k)}), \ldots, (q_N^{(k)}||c_N^{(k)})$ to the previous prompting $P$. However, this approach can quickly reach the maximum input sequence length of the Language Model (e.g. 2048 tokens) and is not particularly scalable or efficient, potentially leading to high levels of redundancy in the prompting and high costs for querying LLM. Therefore, an alternative approach may be needed to address these limitations.

---

[1] The concatenation of question and rationale is denoted as a question-rationale pair $(q||c)$

## 3 PROMPTING OPTIMIZATION FUNCTION

Prompt optimization is a black-box optimization problem, as it can only be evaluated based on the quality of the generated rationales or the correctness of the answers provided by LLM. In this regard, we present our empirical findings on how to design $f$ to update prompting constrained by the limit input sequence length based on two attributes of CoT: **correctness** and **depth**. **Correctness** is a crucial criterion for prompting engineers to update the prompting. Here, we want to ask the question: *Can a valid rationale be replaced with an invalid one without performance drop?* **Depth** refers to the number of reasoning steps which can be reflected by the length of CoT (different steps are separated by comma or \n). A deeper CoT is longer and typically contains more complex rationales, while a shallower CoT is more straightforward and contains fewer reasoning steps. Here, the question is: *Given the similar question, can a deep CoT always be replaced by a shallow CoT?*

## 4 EXPERIMENTS

Our evaluation is conducted on model `text-davinci-002` from OpenAI. We choose multiple datasets across different tasks including arithmetic reasoning (GSM8K Cobbe et al. (2021), Multi-Arith Roy & Roth (2015)), commonsense reasoning (StrategyQA Geva et al. (2021)), and symbolic reasoning (Letter Wei et al. (2022b)). We compare our methods with Zero-Shot-CoT Kojima et al. (2022), and Bootstraping Auto-CoT Zhang et al. (2023a). We manually partition each dataset into 10 batches and generate rationales for each batch under the streaming setting. We report the test accuracy for ten batches. The number of samples in each batch is in Appendix A.

### 4.1 RIGHT OR WRONG CoT

To address this question, we substitute ground-truth rationales that produce the correct answer with Zero-Shot-CoT that produce the wrong answer. During the substitution, we ensure that more than 50% of the rationales in the prompt are incorrect. This substitution method is referred to as *Wrong-CoT*. Conversely, we have *Correct-CoT*, which includes only the correct rationales. Figure 1(a) shows the results. *Wrong-CoT* is manually selected through human evaluation by running the same query multiple times.

### 4.2 DEEP OR SHALLOW CoT

In order to address the aforementioned question, we exclusively utilize CoTs that are deemed correct in the prompt. To distinguish between deep CoT and shallow CoT, we have established a simple heuristic parameter denoted as $\xi$, which is based on the number of \n in the CoT. If the number of \n surpasses $\xi$, we classify it as a deep CoT. Conversely, if a CoT has fewer \n than $\xi$, we classify it as a shallow CoT. *Shallow-CoT* are selected by replacing lengthier rationales by shorter rationales in a batch. Figure 1(b) shows the results.

## 5 CONCLUSION

This paper presents a straightforward case study on how to update and replace prompts used for large language model in a streaming batch setting. Our findings indicate that incorrect chain-of-thought promptings can be valuable and do not significantly diminish performance. Additionally, promptings that consist of shorter rationales exhibit superior performance compared to those with lengthier rationales.

## URM STATEMENT

The authors acknowledge that at least one key author of this work meets the URM criteria of ICLR 2023 Tiny Papers Track.

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

## A  APPENDIX

Table 1: Batch size of four different datasets

|  | **MultiArith** | **GSM8K** | **StrategyQA** | **Letter** |
|---|---|---|---|---|
| **Batch Size** | 60 | 64 | 32 | 81 |

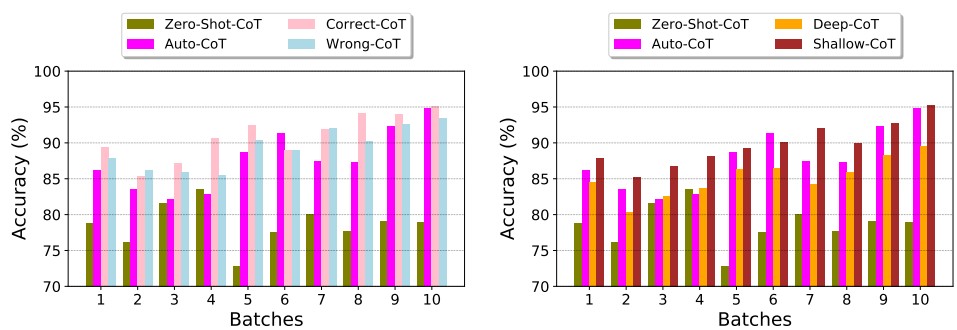

Figure 1: **Left:** accuracy for MultiArith dataset under Correct-CoT and Wrong-CoT. **Right:** accuracy for MultiArith dataset under Deep-CoT and Shallow-CoT with $\xi = 3$.

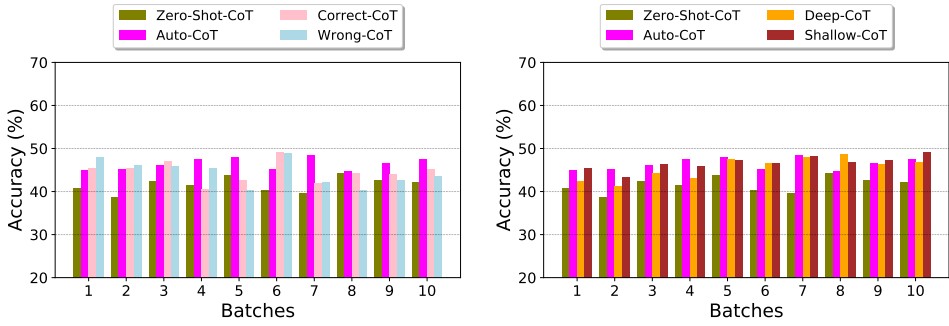

Figure 2: **Left:** accuracy for GSM8K dataset under Correct-CoT and Wrong-CoT. **Right:** accuracy for GSM8K dataset under Deep-CoT and Shallow-CoT with $\xi = 3$.

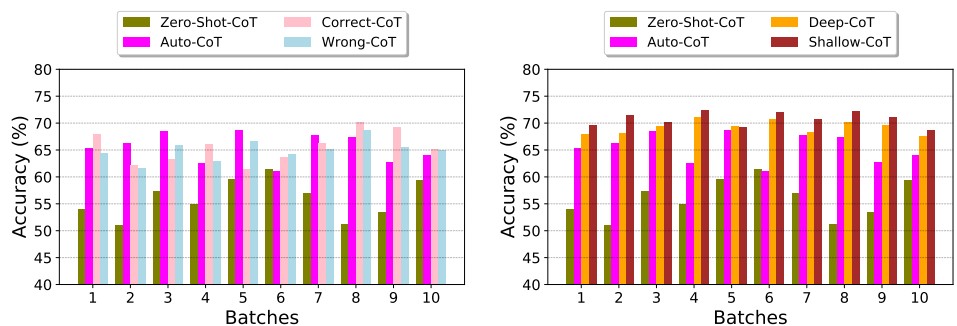

Figure 3: **Left:** accuracy for StrategyQA dataset under Correct-CoT and Wrong-CoT. **Right:** accuracy for StrategyQA dataset under Deep-CoT and Shallow-CoT with $\xi = 3$.

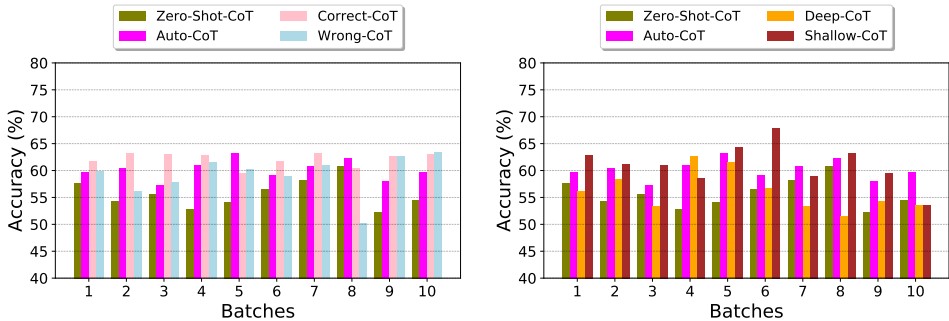

Figure 4: **Left:** accuracy for Letter dataset under Correct-CoT and Wrong-CoT. **Right:** accuracy for Letter dataset under Deep-CoT and Shallow-CoT with $\xi = 4$.

