# OpenReview forum: "Chain Of Thought Prompting Under Streaming Batch: A Case Study"
_ICLR.cc/2023/TinyPapers — Submitted to Tiny Papers @ ICLR 2023_

### Official Review · Reviewer_GJzu · 2023-03-29

**Confidence:** 4

**Summary Of Contributions:**

This paper presents a case study that examines the impact of rationale correctness and depth on the performance of CoT in a streaming batch setting. To assess correctness, the authors replace the gold rationales with incorrect ones produced by Zero-Shot-CoT. To assess depth, the authors use the number of \n to differentiate between deep CoT and shallow CoT.

**Rating:**

Great Start (GS): a submission which meets some of the reviewing criteria but has room for improvement

**Strengths And Weaknesses:**

Strengths

1. Present a case study on how to construct prompts in a streaming batch setting which is useful for other research in this area.

2. The experiments are thoroughly conducted on various datasets.

Weaknesses

1. Some experimental details are missing, and the presentation can be further improved.  Due to space limitations, it is understandable, but I would like to see some explanation on the following aspects:

(1) In section 4.1, how to ensure that more than 50% of the rationales are incorrect?

(2)	How to manually partition the dataset into batches? Do you do it randomly?

(3)	In section 4.2, how to partition an original batch into deep CoT and shallow CoT? If so, I do not think the comparison is fair, since both Zero-shot-CoT and Auto-CoT are experimented on the full original batch.

(4)	In figure 1(right), according to the paper “Automatic chain of thought prompting in large language models”, in the first batch, the Auto-CoT and Zero-Shot-CoT should obtain equal accuracy since they are actually the same in the first batch. Why figure 1 shows a quite different case?

2. The claim is not very well supported.

The paper mentioned that ”Additionally, promptings that consist of shorter rationales exhibit superior performance compared to those with lengthier rationales.”  However, I speculate that for more difficult questions, the rationale may be longer due to the need for additional reasoning steps. Therefore, how can the authors prove that the impact on overall accuracy is due to the depth of the rationale rather than the level of difficulty of the question itself?


**Suggested Changes:**

1. Some necessary explanations can be added. Please see the details in the Strengths And Weaknesses section.

2. Section 4.2 can be further improved to make the experimental design more reasonable and solid.

---

### Official Review · Reviewer_tfdF · 2023-03-29

**Confidence:** 3

**Summary Of Contributions:**

The paper proposes an analysis of chain-of-thoughts for Language Models. Correct and deep chains of thoughts are replaced with wrong and shallow ones, showing a limited loss in performance.

**Rating:**

High Potential (HP): a submission which meets the reviewing criteria and has potential to make an impact on the field

**Strengths And Weaknesses:**

STRENGTHS
==========
I enjoy this kind of analysis, where intuitive principles and general beliefs are questioned. I think the paper touches on important topics, given the recent advancements in Large Language Models.

WEAKNESSES
==========
- I enjoyed the introduction even if I am not an NLP expert, but adding a bit more context and application of the proposed method would be good. Why do we want to replace correct with incorrect CoT? From my understanding, this could lead to faster predictions. If it is the case, can we measure such a gain? I think answering these questions would strengthen the paper's contribution.
- Of course, it is outside the paper's scope, but it is natural to wonder if the findings hold regardless of the language and the underlying model. Maybe this aspect can be listed in the work limitations/future works.

Minors:
- y-axis of the plots does not start from 0. I see that in this way, differences are clear, but it also takes a moment for the reader to interpret the results correctly. I would suggest highlighting this fact by starting the y-axis from 0 and breaking the line (e.g., with a dashed line), or by replacing the bar plots with tables.
- The text refers to Figure 1(a) and Figure 1(b), which is, of course, clear ((a) is left and (b) is right), but it might be good to add these labels to the graphs

**Suggested Changes:**

To improve the submission, I suggest the following changes:
1) Discuss the applicative case, and if possible, quantify the advantage (if any) of Shallow and Wrong CoTs.
2) Mention limitations and future works, pointing out which could be possible to achieve starting from the findings of the paper
3) Modify the graphical items as mentioned above

---

### Official Review · Reviewer_Cu4F · 2023-04-01

**Confidence:** 5

**Summary Of Contributions:**

This paper proposes a case study of Chain-of-Thoughts (CoT) prompting in Large Language Models (LLMs) for batch data in streaming settings. The authors investigated the correctness and the depth of CoT in batch settings.

**Rating:**

Great Start (GS): a submission which meets some of the reviewing criteria but has room for improvement

**Strengths And Weaknesses:**

Weakness
1. "we substitute ground-truth rationales that produce the correct answer with
Zero-Shot-CoT that produces the wrong answer."

How did the author design the wrong answer and the wrong rationales?
How did the wrong answers deviate from the right answers?

(The wrong answer generation can be viewed from a relevant NLP task perspective-- automated multiple choices questions (MCQ) generation. Generating good distractors for MCQ is a challenging and authors first provide heuristics to generate difficult distractors)

2. The authors did not share the source codes or sample prompts.

Strength
The Shallow vs Deep CoT is promising. The Shallow CoT demonstrated better accuracy than the Deep CoT in several batches for all four datasets.

**Suggested Changes:**

1. Please provide how the wrong answers were generated.
2. Please provide sample wrong answer prompts.

---

### Meta-Review · Area_Chair_7cPN · 2023-04-05

**Recommendation:** Invite to present
**Confidence:** 5

**Metareview:**

All reviewers agree that this work studies a promising direction. The analysis also questions intuitive principles and general beliefs that will be of great interest to the NLP community. The writing quality is generally good.

The major concern is the clarity of the experimental setting. The authors may consider improving the paper following the reviewers' suggestions.



**Summary:**

The paper analyzes CoT prompting in LLMs for batch data in streaming settings. Reviewers noted the importance of the paper's contribution to the field and suggested improvements such as providing more context and application of the proposed method, discussing limitations and future works, and addressing missing experimental details.

**Comments And Feedback To The Authors:**

This paper presents an interesting analysis to investigate the impact of the correctness and depth of CoT in batch settings on the overall performance of the model. However, it lacks clarity in the experimental setting and I suggest the authors addressing the following concerns in the next revision.

**(1) Some experimental details are missing:** As pointed out by reviewer 1 and reviewer 3, some necessary explanations are missing including the generation of wrong rationales, the method for partitioning the dataset and sampling prompts.

**(2) Presentation can be further improved**  As mentioned by reviewer 2,  modifications of the graphical items are needed

**Reason For Not Giving A Higher Recommendation:**

N/A

**Reason For Not Giving A Lower Recommendation:**

The paper presents a comprehensive case study that investigates the impact of rationale correctness and depth on the performance of CoT in a streaming batch setting, which is valuable for further research in this field. While the paper lacks clarity and correctness, the author can significantly improve it by providing more experimental details and offering strong experimental results and supporting reasons for their claims. By doing so, the clarity and correctness of the article will be greatly enhanced, leading to more solid results. With these revisions, the paper can be both informative and solid.

---

### Decision · Program_Chairs · 2023-04-07

Invite to present